# Critical Evaluation of Two Commercial Biocontrol Agents for Their Efficacy against *B. cinerea* under In Vitro and In Vivo Conditions in Relation to Different Abiotic Factors

Gurkan Tut [1,2], Naresh Magan [2,*], Philp Brain [1] and Xiangming Xu [1,*]

1 NIAB East Malling Research, West Malling, Kent ME19 6BJ, UK; Gurkan.Tut@hotmail.com (G.T.); Philip.Brain@NIAB.com (P.B.)
2 Applied Mycology Group, Environment and AgriFood Theme, Cranfield University, Cranfield, Bedford MK43 0AL, UK
* Correspondence: n.magan@cranfield.ac.uk (N.M.); Xiangming.Xu@niab.com (X.X.); Tel.: +44-1234-758308 (N.M.)

**Abstract:** The study evaluated the dose–response relationship of two commercial microbial biocontrol agents, *Bacillus subtilis* and *Gliocladium catenulatum*, against *Botrytis cinerea* both in vitro and in vivo. Inoculum doses, formulation, temperature and foliar leaf part all affected the control achieved by the two BCAs. In vitro competition assays on modified PDA plates tested a range of BCA doses ($\log_{10}$ 3–10 CFUs or spores/droplet) at 4, 10 and 20 °C on the development of *B. cinerea* colonies. The dose–response relationship was influenced by both the BCA formulation and temperature. In vivo studies on lettuce plants in semi-commercial greenhouses examined the BCA dose ($\log_{10}$ 5–9 CFUs or spores/mL) for controlling *B. cinerea* with a high inoculum ($\log_{10}$ 6 spores/mL). Leaf disc assays showed that the dose–response relationship was influenced by the leaf parts sampled. These results suggest that the dose–response relationship between a BCA and specific pathogen will be significantly influenced by environmental conditions, formulation and plant phyllosplane tissue.

**Keywords:** commercial biocontrol agents; formulation; temperature; *Botrytis cinerea*; *Bacillus subtilis*; *Gliocladium catenulatum*; biocontrol; dose–response; spatial effects

## 1. Introduction

*Bacillus subtilis*, a Gram-positive *Rhizobacterium* produces endospores for survival and lipo-peptides (iturins, fengycins, surfactins) for plant colonization, induction of plant defense responses and control of plant pathogens [1]. The ability of *B. subtilis* to compete for space and nutrients is important for the survival and colonization of plants [2]. Serenade ASO® (Bayer Crop Science), a globally used broad spectrum bio-fungicide, contains *B. subtilis* strain QST 713 and is registered in Europe to manage *Botrytis cinerea* on crops, including lettuce and strawberry [3]. *Gliocladium catenulatum*, a saprophytic filamentous fungus, survives on organic matter and as an endophyte in roots and stems, is rhizosphere-competent and reportedly competitive against a range of fungal plant pathogens [4]. *G. catenulatum*, which has also been suggested to be parasitic to fungal pathogens [5], destroying hyphal cells [6], produces enzymes (chitinase, β-1,3-glucanases) for the hydrolysis of fungal cell walls [7] and is effective at competing for space and nutrients in plants [8]. Prestop® (ICl Ltd., Ipswich, Suffolk, UK), broad-spectrum bio-fungicide, contains conidia and mycelium of *G. catenulatum* strain J1446 and is registered in Europe to manage *B. cinerea* on crops.

*B. cinerea* is an economically important pathogen of lettuce. Under conducive conditions, conidia infects the base of the plant and the disease develops upwards throughout the head until all inner leaves are altered into a slimy mass [9,10]. The pathogen rots the stem and leaves at the center before reaching the margin of the outer leaves. Characteristic ashen-grey lesions form with conidial sporulation just above the surface of the diseased

area. They are often on the underside of the leaves where the moist microclimate is better. Subsequently, sclerotia develop throughout the decayed leaves [9,10].

Serenade ASO® is a suspension concentrate containing of *Bacillus amyloliquefaciens* (=*subtilis*) strain of QST 713 (including fermentation residues and water with a minimum of $1.05 \times 10^{12}$ CFUs/L. PRESTOP® is a biofungicide consisting of a 32% *w/w* wettable powder containing a nominal $2 \times 10^8$ CFUs/g and applied at the rate of 500 g per 100 L (0.5%). For these foliar applications, a better understanding of the BCA-pathogen dose relationships are needed before such bio-fungicides can be effectively exploited to manage more generally plant diseases in commercial agriculture and horticulture.

Generally, dose–response relationships are often characterised by probit models. For determining timing of BCA applications, considerations include pathogen infection risks, ensuring viable biocontrol population sizes and the BCA/pathogen dose relationship under specific environmental conditions [11]. Surprisingly, there has been no published information on the dose–response model relationships for *B. subtilis* and *G. catenulatum* against *B. cinerea*. These models would be very valuable to obtain information on the effective dose required for reduction of the pathogen population by 50% ($LD_{50}$). In addition, the efficacies will be influenced by both biotic and abiotic factors, including the formulation and target host tissue. Thus, intra-leaf positions and target leaf parts are a critical step in understanding the ecology of the BCA and pathogen. Such information would enable a more targeted approach to maximise the application timing and targeting, especially of phylloplane surfaces, to optimise biocontrol efficacy [12].

The objective of the present research was to evaluate the dose–response relationships of these two BCAs (*B. subtilis* QST 713, *G. catenulatum* J1446) against the pathogen *B. cinerea* (a) in vitro and (b) in-situ in relation to temperature and BCA formulation, as well as biotic spatial effects in the phylloplane of lettuce leaves.

## 2. Materials and Methods

### 2.1. Isolation and Culture Conditions of B. subtilis QST 713, G. catenulatum J1446 and B. cinerea

Serenade® ASO (Serenade) and Prestop were purchased from Fargro Ltd (Arundel, West Sussex, UK). Serenade was an aqueous formulation and contained *B. subtilis* strain QST 713. Prestop® was a dry formulation and contained *G. catenulatum* strain J1446. For both bio-fungicides, culture conditions, growth and formulation components were unknown due to industrial proprietary. Serenade was stored at room temperature, while Prestop was stored in a cool dry location at <8 °C and, once opened, frozen at −20 °C. The batches used were less than 6 months old. It should be noted that up to $\log_{10}$ 1 CFUs or conidia/mL were found to be non-viable in the formulations when counted in a haemocytometer and plated on agar media. Each BCA was isolated from their formulations and grown in vitro on culture media. Serenade was serially diluted thrice into maximum recovery diluent (Sigma Aldrich, Gillinham, Dorset, UK) and 10 μL was spread-plated onto nutrient agar (NA, Oxoid Ltd) and incubated at 30 °C for three days. Concentrates were produced by collecting the bacterial colonies on the media into maximum recovery diluent solution and transferring the supernatant (*B. subtilis* suspended into maximum recovery diluent) onto the next plate to repeat the process. The concentrate was stored at 20 °C. One gram of Prestop was mixed with 200 mL of maximum recovery diluent, shaken vigorously for 15 s and serially diluted twice, before 10 μL were plated. Then, the mixture was spread-plated on malt extract agar (MEA, Oxoid Ltd) and incubated at 22 °C for 10 days. Concentrates were produced by collecting the surface fungal growth into maximum recovery diluent solution and transferring the supernatant (*G. catenulatum* suspended into maximum recovery diluent) to the next plate and repeating the process. The hyphae and mycelia were separated from the macroconidia in the suspension by filtration (Whatman 25 μm).

For in vitro dual culture assays, three single-spored *B. cinerea* strains were isolated from separate strawberry fruits (cultivar Elsanta) with grey mould symptoms at a strawberry production facility at East Malling and plated on potato dextrose agar (PDA, Oxoid

Ltd), followed by incubation at 20 °C for 10 days in the dark. These isolates were used because these studies were performed out of lettuce production season. The fungal colonies from each plate were placed into the same sterile beaker, mixed with maximum recovery diluent solution and stored at 4 °C. From the mixed isolate suspension, 200 µL was spread-plated onto PDA, followed by incubation at 20 °C for 10 days in the dark. For in planta experiments, four isolates of *B. cinerea* were obtained directly from lettuce leaves (cvs. Cos Romia, Little Gem, Lollo Verdi and Apolo) infected with *B. cinerea* from a commercial lettuce field in West Malling, Kent and three further *B. cinerea* isolates were collected from lettuce (cv. Carter) infected with the pathogen from a semi-commercial glasshouse at NIAB EMR. The collected isolates were cultured on PDA and incubated at 20 °C for 10 days with 8 h light/16 h dark cycles. For acquiring conidia from the *B. cinerea* cultures, 10 mL of sterile distilled water was added to the colony surface and the culture agitated for conidial release. Supernatants were collected, filtered (Whatman 25 µm), serially diluted and confirmed with microscopic counts using a haemocytometer, to contain macro-conidial concentrations of about $1 \times 10^6$ spores/mL. The in vitro experiment used a mixture of all three isolates from Carter, while the in-planta experiment used a mixture of all the lettuce isolates combined as a mixed pathogen *B. cinerea* inoculum.

### 2.2. In Vitro Dual Culture Co-Inoculations

The viable plate count technique was used to confirm BCA dosage (*B. subtilis*, *G. catenulatum*). For Serenade (*B. subtilis*), assays were performed on mixed media consisting of 50% of the recommended PDA and NA agars, while for Prestop (*G. catenulatum*), assays were performed on 50% of recommended PDA and 50% of the MEA. For non-formulated strains, the concentrations used were between $5 \times 10^3$ and $5 \times 10^9$ (at an increment of one order of magnitude) CFUs for *B. subtilis* and spores for *G. catenulatum* per droplet. For the two formulated products, eight inoculum doses were tested, ranging from $5 \times 10^3$ to $5 \times 10^{10}$ (at an increment of one order of magnitude) per droplet in the formulated product.

Testing was carried out in single vent 90 mm × 16.2 mm Petri plates. A modified dual culture technique [13] was used, in which an agar plug of 34 mm in diameter was removed with a surface sterilised cork borer and then re-plugged with a mycelial plug of the same size containing both the agar and the *B. cinerea* fungal layer. In succession the BCA droplet was applied. The plug site of the pathogen and the application site of the BCA droplet in the Petri plate were opposite each other, 6.5 cm apart and 1 cm from the edge of the plate. The plates were sealed with parafilm and incubated at 4, 10 or 20 °C for 7 days in the dark. The positive control contained the *B. cinerea* mycelial plug only and the negative control contained no microorganisms. After 7 days, images of dual cultures and positive controls were obtained and analysed with ImageJ (National Institutes of Health, Bethesda, MD, USA) to calculate the *B. cinerea* mycelial area colonised. Each treatment (BCA × dose × temperature) contained a maximum of thirty and a minimum of nine replicates, these replicates were broken down into three independent experiments using a randomised block design.

### 2.3. Effect of the Two BCAs on Control of B. cinerea on Lettuce Leaves

Lettuce cultivar Carter was grown in pots (9 × 9 × 10 cm) with Miracle-Gro All Purpose Premium Compost (Evergreen Garden Care Ltd., Frimley, Surrey, UK) and placed in a semi-commercial pest- and disease-free glasshouse until early head development with about 4–6 bottom leaves. During experimentation, lettuce plants were hand watered daily and the glasshouse temperature and relative humidity ranged between 17 and 22 °C and 60% and 95%, respectively.

The viable plate count technique was used to confirm the BCA dosage. At the point of early head development, the lettuce plants were spray-inoculated with a hand sprayer in fine droplets and sprayed until just before run-off with the individual BCA or with sterile distilled water for the negative and positive controls. Five doses tested were $5 \times 10^5$, $5 \times 10^6$, $5 \times 10^7$, $5 \times 10^8$ and $5 \times 10^9$ CFU/mL for Serenade and spores/mL for Prestop.

Four hours after the BCA application, plants were similarly spray inoculated with *B. cinerea* (macro-conidial dosage of $1 \times 10^6$ spores/mL), except for the negative control which was sprayed with sterile distilled water. For 48 h, the glasshouse ventilation windows were closed to encourage infection and after 48 h, two older leaves from each plant (defined by their larger size and marked with a permanent marker at the time of inoculation) of similar size and shape were collected and surface sterilized. Phylloplane leaf surface washing consisted of initially using slow running cold tap water, followed by a 1 min wash in tween 80 solution (1 drop in 200 mL sterile distilled water), re-washed with sterile distilled water twice, then surface sterilized with a wash in 70% ethanol for 1 min. The residual alcohol was removed by washing twice in sterile distilled water, followed by leaf drying under a fume hood for 2 h. From each older lettuce leaf, a total of ten leaf discs (10 mm diameter) were obtained and leaf parts included the apex (1 disc), midrib (3 discs) and lamina with lateral veins (6 discs). Figure 1 illustrates the relative positions of the 10 sampled leaf disc on each leaf. The discs were placed on PDA at an equal distance from each other (0.75 cm), the plate was sealed with parafilm and incubated at 20 °C for 10 days in the dark. After the incubation period, the leaf discs were assessed for the incidence of disease and/or symptoms (i.e., lesions, hyphae and necrosis). The experiment was performed twice. In each experiment for each treatment (BCA × dose), there were five biological replicates (five plants), and from each plant two of the oldest leaves were obtained for experimentation (two replicates). Therefore, each treatment dose contained a total of twenty leaves. The testing followed a randomised block design.

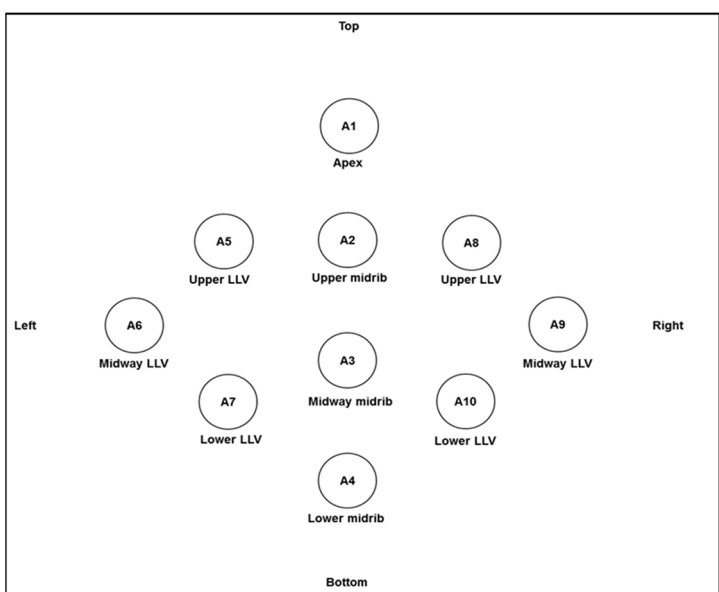

**Figure 1.** Diagrammatic representation of leaf disc positions taken from older *L. sativa* leaves (This is not to scale. Key: LLV, lamina and lateral veins.

### 2.4. Data Analyses

Generalised linear models (GLM) were used to analyse the dose–response relationship, assuming residual errors to follow a binomial distribution. For in vitro dual culture data, biocontrol efficacy (E) was calculated as:

$$E = [(C - T)/C] \times 100$$

where C and T was the area of *B. cinerea* mycelial colonisation of the agar plate for the control and treatment, respectively. In GLM analysis of the efficacy data for each BCA, formulation (formulated or not) and temperature (4, 10 and 20 °C) were treated as factors. Any E values less than 0 were set to 0, and similarly those values greater than 100 were set

to 100. In GLM analysis of the efficacy (E) data, the total counts for individual data points was assumed to be 100.

For the leaf disc data, GLM was applied to the data at two levels. At the first level, data were pooled over all leaf disc positions from the two repeat experiments. Percent leaf discs infected for each combination of BCA and dose was first adjusted for the level of latent (pre-existing) infection of *B. cinerea* (as in the negative control (NC)) and the maximum level of infection (as in the positive control (PC)), namely:

$$\text{Iadj} = (\text{I} - \text{INC})/(\text{IPC} - \text{INC}) * 100$$

where I and Iadj are respectively observed and adjusted percent infection for a treatment, and IPC and INC are observed percent infection for the NC and PC treatments, respectively. In GLM analysis, for each combination of BCA and dose, the total leaf disc was 200 and the number of infected leaf disc was estimated from Iadj. BCA was included in GLM as a factor.

At the second level, GLM was applied to data at individual leaf positions. Percent infection was similarly adjusted for specific leaf locations for each BCA. Any adjusted values less than 0 were set to 0, and similarly those values greater than 100 were set to 100. In GLM analysis, for each combination of BCA, leaf position and dose, the total number of leaf discs was 20 and the number of infected discs was estimated from Iadj. Leaf position and BCA were treated as factors in GLM.

For all GLM analyses of the data, an additional data point at zero dose with zero efficacy was added. Accumulated deviance analysis was used to assess the effects of treatment factors on the dose–response relationship. All analyses were carried out in R (version 4.1.0).

## 3. Results

### 3.1. Effect of BCA Inoculum Dose, Formulation and Temperature on In Vitro Control of B. cinerea

Figures 2 and 3 show that the relative inhibition of *B. cinerea* mycelial colonisation occurred in all tested temperatures with both the BCAs and was generally increasing with dosage. The highest inoculum doses assayed with Serenade and Prestop resulted in the highest control efficacy of 96% and 91%, respectively.

For the *Bacillus*, analysis of accumulated deviance showed that both formulation and temperature affected ($p < 0.001$) the dose–response relationship, affecting both the intercept and slope parameters. Formulated strains retained greater efficacy at each temperature and inoculum dose (Figure 2), particularly for low doses. Indeed, the higher-than-expected efficacies at low doses for the formulated product led to large deviations in model predictions.

Similarly, for the *Gliocladium*, both temperature and formulation affected ($p < 0.001$) the observed dose–response relationship. The effect of dose was less profound at 10 °C for the unformulated strain (Figure 3).

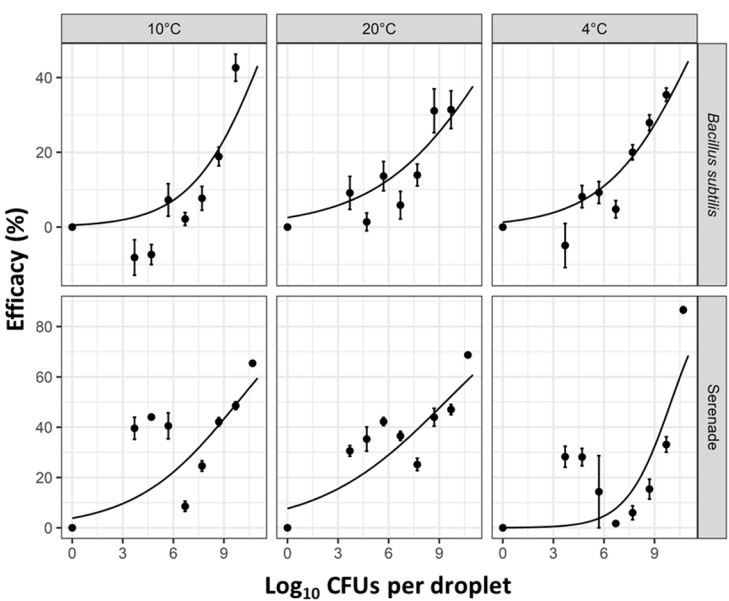

**Figure 2.** Efficacy from in vitro competition assays of *B. subtilis* and Serenade at tested inoculum doses inhibiting growth from a *B. cinerea* mycelial plug at 4, 10 and 20 °C; the bars represent one standard error and the curve is a fitted logistic model.

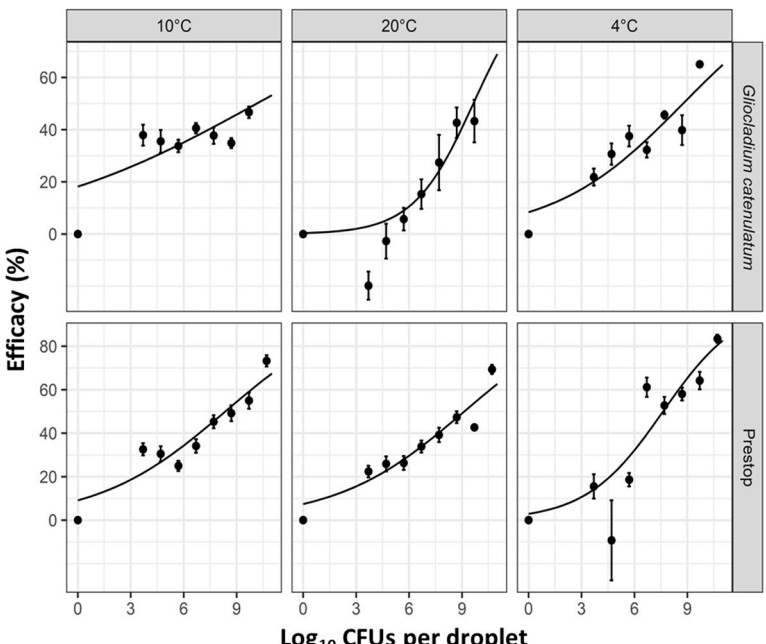

**Figure 3.** Efficacy from in vitro competition assays of *G. catenulatum* and Prestop at tested inoculum doses inhibiting growth of a *B. cinerea* mycelial plug at 4, 10 and 20 °C; the bars represent one standard error and the curve is a fitted logistic model.

*3.2. In Vivo Dose–Response Relationship and Inoculum Dynamics of the BCAs on Lactuca Sativa Leaves under HIgh B. cinerea Inoculum Pressure*

3.2.1. Analysis of the Leaf-Disc Data Ignoring the Spatial Location

Figure 4 shows the incidence of leaf discs infected with *B. cinerea* in relation to the application of Serenade and Prestop at several doses following inoculation with *B. cinerea*. As BCA inoculum dose increased, the infection level decreased. Complete or close to complete control (100%) of *B. cinerea* may be achievable with the highest inoculum dose of $\log_{10}$ 9.7 CFUs or spores/mL.

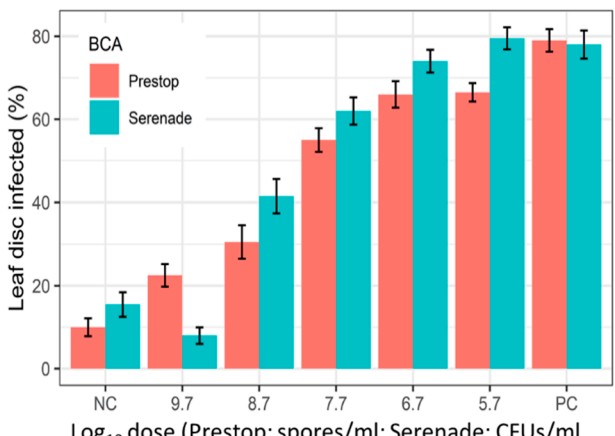

**Figure 4.** Overall percent *L. sativa* leaf disc infected with *B. cinerea* when post-inoculation challenged with two formulated biocontrol products (Prestop and Serenade) at a number of doses in a glasshouse at 17–22 °C with a RH range of 60–100%. NC: negative control (only sterile water was applied) and PC: positive control (inoculated with *B. cinerea* but not treated with BCA).

Logistic models satisfactorily described the observed dose–response relationships (Figure 5). The response curve is steeper ($p < 0.05$) for Serenade than for Prestop. However, $LD_{50}$ for both BCAs were close, at around $\log_{10}$ 8.0 CFUs or spores/mL (see Figure 5).

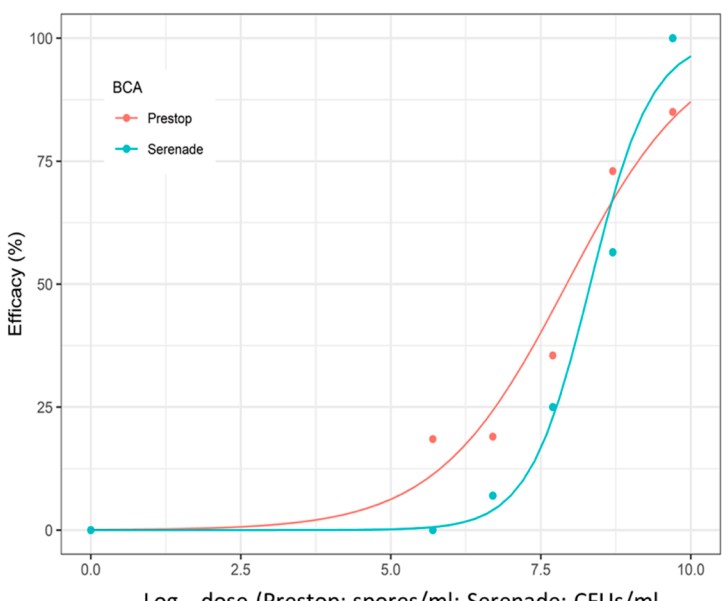

**Figure 5.** Fitted logistic models describing the dose–response relationship of *B. cinerea* infection of lettuce leaves in relation to different doses of the two biocontrol (BCA) agents (Prestop and Serenade) examined.

### 3.2.2. Effect of Spatial Leaf Disc Position on Dose–Response Relationship of the Two BCAs for *B. cinerea* Control

Analysis of accumulated deviance indicated that both inoculum dose and leaf disc position affected ($p < 0.001$) the control efficacy achieved by two BCAs. These two factors affected both the intercept and slope parameters. Figure 6 shows the observed efficacy data as well as the fitted models for each leaf disc position.

For Prestop, the difference in the dose–response relationship among 10 leaf positions resulted mostly from the fact that positions A4 and A6 differed greatly from other positions, with much lower efficacies achieved (see Figure 6), which was confirmed by the analysis of

deviance. For Serenade, the dose–response relationship for position A6 differed largely from other positions, with a much steeper response. Only the two highest doses led to disease control (see Figure 6), and the significance of this difference was indicated by the analysis of deviance.

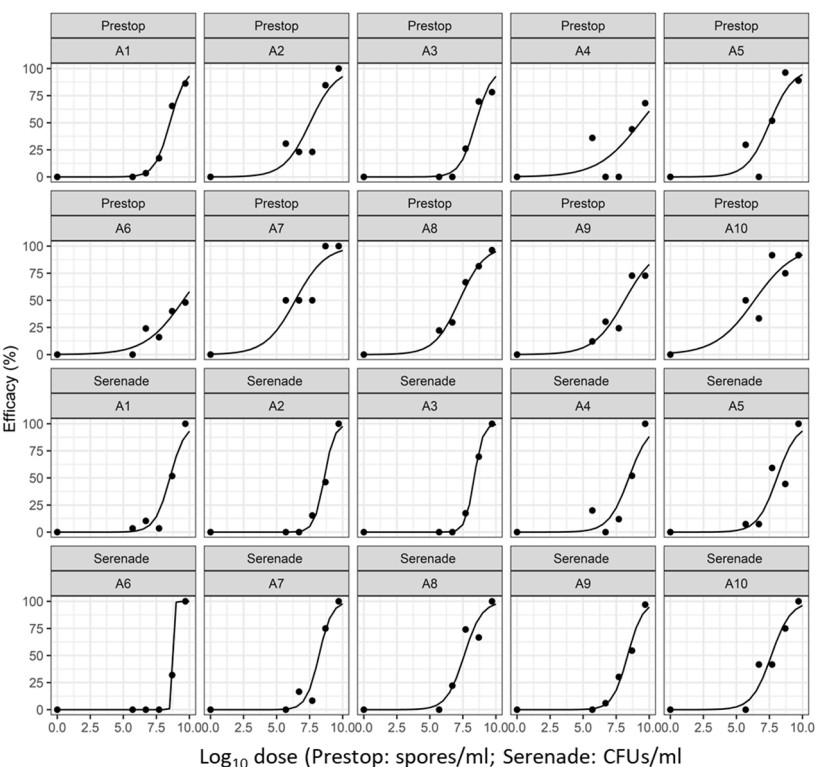

**Figure 6.** The effect of leaf position on the biocontrol efficacy on *L. sativa* leaf discs from attached lettuce leaves pre-inoculated with *B. cinerea* and then challenged with two biocontrol products (Prestop and Serenade) at different doses in a glasshouse at 17–22 °C with a RH range of 60–100%. Curves are fitted logistic models describing the observed dose-response relationships. The exact leaf positions are given in Figure 1.

## 4. Discussion

This study has examined the dose–response relationship between the BCAs *B. subtilis* and *G. catenulatum* and the control of *B. cinerea* applied initially at a high inoculum both in vitro and in vivo. This has allowed the development of dose–response relationships of the two BCAs against *B. cinerea* on different phylloplane leaf regions.

In vitro studies with both BCAs reduced mycelial growth of the *B. cinerea* in culture. Interestingly, the formulated BCAs performed better (particularly at low doses) and had higher efficacies than the unformulated microorganisms. This may be partially due to proprietary additives, including stickers and adjuvants, which may improve establishment, especially on phylloplane surfaces. Temperature impacted on the dose–response relationship of the two BCAs. Temperature influences reproduction of both BCAs and the synthesis of lipo-peptides in *B. subtilis* and hydrolytic enzymes in *G. catenulatum*. *B. subtilis* growth becomes marginal at approximately 11 °C, and cold shock proteins, fatty acids and SigB proteins are produced for cellular survival [14], which provides some resilience to environmental stress [15]. Lipo-peptides produced by *B. subtilis* are involved in antifungal activity, biofilm formation and colonization. Examples include iturin over production in solid-state fermentation at lower temperatures [16], surfactin overproduction at 37 °C [17] and an increase in mycosubtilin production when temperature decreases from 37 °C to 25 °C [18]. *G. catenulatum* can grow over a wide temperature range (5–34 °C) and survives at up to 42 °C. Optimal growth occurs at 15–25 °C [4]. Knowledge of the

ecological windows for the competitiveness of the BCA is important and these windows need to mirror those of the pathogen phase of interest to maximize control [19]. Thus, the effect of interacting conditions of water availability–temperature influences the biomass production of *Gliocladium* species [20]. This species is able to produce heat shock and cold shock proteins which improve the stress tolerance [21]. Hydrolytic enzymes produced by *G. catenulatum* are known to be involved in antifungal activity. This BCA species has been shown to produce β-1,3-glucanase and chitinase [6], and the stability of chitinase was up to 40 °C, contributing to efficacy against *B. cinerea* [22]. In addition, this species is able to produce a perilipin protein encoded by the *Per3* gene which is involved in enhanced myco-parasitic activity at 28 °C against sclerotia [23]. The production of cold shock proteins in both BCAs, in addition to the synthesis of hydrolytic enzymes and mycoparasitic proteins by *G. catenulatum*, as well as the production of SigB proteins and lipo-peptides by *B. subtilis*, probably all contribute to the control of *B. cinerea* colonization of in the phyllosphere of different horticultural crops [24].

This study has shown that these two BCAs prevented *B. cinerea* conidial activity at high inoculum doses on older lettuce leaves. The maximum mean efficacy of the BCAs on older leaves were obtained at the highest dose applied at $5 \times 10^9$ CFUs/mL, which is much higher than the recommended dose (approximately $10^7$–$10^8$ CFUs/mL). For Serenade, this highest dose achieved 92% and was greater compared to grey mould disease of apples with *B. subtilis* GA1 (80%) at $5 \times 10^8$ [22] and *Botrytis* blight of geranium with *B. subtilis* QST 713 (Serenade max) [23]. Meanwhile, Prestop achieved an efficacy of 77% (at around $10^6$ CFUs/mL) which was greater than that reported for *Botrytis* blight of geranium with Prestop [25] and similar to *B. cinerea* stem infection in cucumber and tomato with *G. catenulatum* (>75 %) [26]. The $LD_{50}$ was around $\log_{10}$ 8.0 for Serenade (CFUs/mL) and Prestop (spores/mL) for preventing a high inoculum of *B. cinerea* ($\log_{10}$ 6 spores/mL) on older lettuce leaves.

In the present study, the dose–response relationship was significantly affected by the exact leaf disc position for both BCAs, particularly in the mid lamina and lateral veins with much lower efficacies. It is possible that the spatial position of the leaf parts affected the BCAs and *B. cinerea*, including their retention and survival, because of differences in the chemical and physical properties of these phyllosphere surfaces. The presence of symptomless systemic *B. cinerea* infection of lettuce initiated from the roots and migrating into the stem, petiole and leaves suggests that *B. cinerea* populations may reside in the vascular tissue in the midrib area, thus requiring a higher BCA concentration to achieve control. This may also explain the positive infections found in some of the negative controls in the midrib leaf sections [27]. There may also be some morphological differences spatially, which may provide better microclimate conditions for *B. cinerea*. However, more information is required concerning these aspects and their relative nutrient contents [12]. Other key features, such as cuticles [28], wound and stomatal sites [29,30] and the capacity for producing cell wall degrading enzymes [31], may all influence the level of control of *B. cinerea* obtained with these two BCAs. They may contribute to the observed decline in the control efficacy as well as the shift of the dose–response curve for both BCAs in the apex and midrib regions in contrast to lamina and lateral veins. The competition for space and nutrients [2], lipo-peptide synthesis [1] and induced host resistance [32], especially for *B. subtilis*, and in addition antifungal enzyme production and hyper parasitism [5,7,23] by *G. catenulatum*, probably also contributed to the control of *B. cinerea* on older lettuce leaves.

The present study has drawn attention to the importance of temperature, formulation type and biotic (leaf parts) factors influencing the dose–response relationships, which provide an effective starting point for optimizing the chances of effective biocontrol. Further studies are needed to study the effect of *B. cinerea* disease pressure/inoculum on dose–response relationships. The approaches developed in this study should help improve the application strategies for these commercially available BCAs.

Temperature influences the reproduction of both BCAs and the synthesis of lipo-peptides in *B. subtilis* and hydrolytic enzymes in *G. catenulatum*. *B. subtilis* growth becomes

marginal at approximately 11 °C, and cold shock proteins, fatty acids and SigB proteins are produced for cellular survival [14], which provides some resilience to environmental stress [13]. Lipo-peptides produced by *B. subtilis* are involved in antifungal activity, biofilm formation and colonization. Examples include iturin overproduction in solid-state fermentation at lower temperatures [16], surfactin overproduction at 37 °C [17] and an increase in mycosubtilin production when temperature decreases from 37 °C to 25 °C [18]. *G. catenulatum* can grow over a wide temperature range (5–34 °C) and survives at up to 42 °C. Optimal growth occurs at 15–25 °C [4]. Knowledge of the ecological windows for the competitiveness of the BCA is important and these windows need to mirror those of the pathogen phase of interest to maximize control [19]. Thus, the effect of interacting conditions of water availability–temperature influences the biomass production of *Gliocladium* species [20]. This species is able to produce heat shock and cold shock proteins, which improves the stress tolerance [21]. Hydrolytic enzymes produced by *G. catenulatum* are known to be involved in antifungal activity. This BCA species has been shown to produce β-1,3-glucanase and chitinase [6], and chitinase was stable at up to 40 °C, contributing to efficacy against *B. cinerea* [22]. In addition, this species is able to produce a perilipin protein encoded by the *Per3* gene, which is involved in enhanced mycoparasitic activity at 28 °C against sclerotia [23]. The production of cold shock proteins in both BCAs, in addition to the synthesis of hydrolytic enzymes and mycoparasitic proteins by *G. catenulatum*, as well as the production of SigB proteins and lipo-peptides by *B. subtilis*, probably all contribute to the control of *B. cinerea* colonization in the phyllosphere of different horticultural crops.

## 5. Conclusions

This study has shown that both BCAs inhibit the colonization of *B. cinerea* inoculum in vitro over a range of temperatures, with formulated versions having an advantage in the control efficacy. Dose–response curves of both BCAS against *B. cinerea* were developed and shown to be influenced by temperature, formulation and leaf part. In situ studies on lettuce leaves, especially the spatial variability of biocontrol related to leaf positions, showed the importance of taking the phyllosphere morphology into account when delivering BCAs to such surfaces to control this important pathogen.

**Author Contributions:** G.T. was a PhD student who executed the research work and prepared the first draft; X.X. and N.M. devised the project and were the PhD supervisors, corrected the drafts, provided additional statistical analyses and revised the final submitted version of the manuscript; P.B. carried out the statistical analyses of the data sets. All authors have read and agreed to the published version of the manuscript.

**Funding:** This research received no external funding.

**Institutional Review Board Statement:** Not applicable.

**Data Availability Statement:** All the data sets for statistical analyses have been deposited with the corresponding authors, X. Xu and N. Magan, and are available from NIAB-EMR or Cranfield University.

**Acknowledgments:** G.T. is grateful to the AHDB for financial support for the PhD project (Project number CP140).

**Conflicts of Interest:** The authors declare no conflict of interest.

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
