# Peer review of "Critical Evaluation of Two Commercial Biocontrol Agents for Their Efficacy against B. cinerea under In Vitro and In Vivo Conditions in Relation to Different Abiotic Factors"

_agronomy, doi:10.3390/agronomy11091868_

Round 1

Reviewer 1 Report

Please examine the edited manuscript, there are several suggestions for improvement.  Furthermore, I have several questions that require the attention of the authors and clarification of methodology etc.

I would like to see a meaningful comparison of the doses used with the field applied rates that are described for these commercial products.

An improved understanding of the key pathogen (Botrytis) and lettuce as a plant.

Author Response

We thank the Referee for the comments on the manuscript. We have tried to address all the changes recommended. These are all now in blue text so they changes made can be observed easily. 

We have included the formulation details and the lables for te concentrations of the two BCAs which are present. However, it should be noted that up to log10 1 CFUs/conidia were non viable in the formulations. 

We have added a box for the sampling of the phylloplane tissues from the leaves instead of drawing a leaf outline. We hope this is acceptable. We have also tried to clarify all the questions posed. 

We hope these have been answered satisfactorily. 

Prof. N. Magan

Reviewer 2 Report

Please, find attached here with my revision report.

Author Response

Title I suggest to change it as “Critical evaluation of two commercial biocontrol agents for their efficacy against B. cinerea under in vitro and in vivo conditions in relation to different abiotic factors”.

Answer: Done

Abstract The authors should better focus the results obtained in the abstract section.

Answer: We think this is relatively succinct and has clarity at the present time on what was found.

Introduction

The authors should give a brief history on the regulatory use and characteristics of the two commercial formulates Serenade and Prestop containing microbial biocontrol agents, Bacillus subtilis and Gliocladium catenulatum, respectively, against Botrytis cinerea on lettuce in the Introduction session. More of them have been reported yet in M&M, they should be moved only. In addition, a brief description of grey mould by B. cinerea on lettuce should be provided in the same section.

Answer: The details of how these were developed are not really relevant here. These are registered commercial products for use with clear labelling and use option for foliar sprays. These have been mentioned in the Introduction and the type of formulations clarified.

We have also included some text about Botrytis infection of lettuce and added some references.

M&M -Why the authors have used different isolates of B. cinerea for in vitro (from strawberries) and in vivo (from lettuce fields) trials? - with a standard compost? Please, specify origin and composition of it. Only compost? I really do not think so. -delete the term “biological” before “replicates”.

Answer: We have clarified these points and made the changes.

Results -L. 202. Change in: Similarly, for the Gliocladium both….

Answer. This has been changed.

Discussion -L. 264. Please remove “for the first time”.

 -L. 268-271. Please quote the sentences with literature.

-L.271-294. I suggest to move this text at the end of the Discussion section as it contains more useful information which, however, do not constitute the main objective of this work.

-L.331-334. I suggest to move this text to the Conclusion session

Answer. All these have been changed

All changes made have been done in blue text for identifying those which have been made.

Prof. N. Magan